# Visible Lymph Affluents in the D3 Volume: An MDCTA Pictorial Essay

**DOI:** 10.3390/diagnostics12102441

**Published:** 2022-10-09

**Authors:** Bojan V. Stimec, Dejan Ignjatovic

**Affiliations:** 1Anatomy Sector, Teaching Unit, Faculty of Medicine, University of Geneva, 1211 Geneva, Switzerland; 2Department of Digestive Surgery, Akershus University Hospital, University of Oslo, 1478 Lorenskog, Norway; 3Institute of Clinical Medicine, University of Oslo, 0315 Oslo, Norway

**Keywords:** complete mesenteric excision, D3 volume, lymphatic vessel, intestinal trunk, multidetector computed tomography angiography, 3D segmentation, chylous ascites, mesentery

## Abstract

Background: There seems to be a gap in knowledge of the anatomy of mesenteric lymphatics between the superior mesenteric nodes and the intestinal trunk. To our knowledge, these central lymph vessels were not hitherto systematically searched for, described, or morphometrically analyzed. Our aim was to identify those vessels on the routine multidetector computerized tomography angiography (MDCTA), performed prior to right colectomy for cancer, with extended mesenterectomy, central vascular ligation, and D3 lymphadenectomy. Methods: A total of 420 MDCTA datasets were analyzed utilizing manual segmentation and 3D reconstruction, with the aid of image processing software Osirix, Mimics, and 3-matic. The 3D models and masks underwent a detailed topographic and morphometric analysis. Results: Significant vascular-like structures, having neither origin nor termination on the blood vessels, were noted in 18 cases (4.3%) in the D3 volume. The dimensions of visible lymph vessels varied, their mean diameter was 1.81 ± 0.61 mm, and the mean length was 38.07 ± 22.19 mm. In the vast majority of cases, the lymph vessels were situated in front of the superior mesenteric artery (SMA), coursing either longitudinally cranially (13 cases) or transversely/obliquely to the left (5 cases). In all cases but one, the lymph vessel passed at the left-hand side of the middle colic artery. As for the course shape, in seven cases, the lymph vessel appeared highly serpiginous. Conclusions: The regular MDCTA can provide valuable information on mesenteric lymphatics and aid in surgical planning.

## 1. Introduction

The lymphatic system has sometimes been addressed as one of the most complicated of *Homo sapiens* [1]. Its anatomy and physiology have been less appreciated than the ones of its “older brother”, i.e., of the blood vessels [2]. There are many issues concerning the lymphatic system which are still under scientific scrutiny, e.g., its embryology [3]. As for the anatomy, it is conventional to classify the lymphatic network into capillaries, larger lymph vessels, lymph nodes, and large lymph vessels, the latter responsible for draining the lymph into the venous system [1].

### 1.1. Literature Review 

The more and more accepted complete mesocolic excision with D3 lymphadenectomy for cancer [4,5], either by open access, laparoscopy or robotic surgery has shed new light on the lymphatic anatomy of the mesentery of the small and the large bowel. The D3 volume has been defined according to the landmarks of the superior mesenteric vein (SMV) and artery (SMA) and their principal affluents/branches [6,7]. The limits are as follows: 2 cm-distal to the origin of the ileocolic artery and cranial to the origin of the middle colic artery (MCA), 3 cm right to the right border of the SMV, and the left border of the SMA. This volume includes the tissue in front and behind the mesenteric vessels, with the posterior limit of the fusion fascias of Treitz and Toldt. It is well known that the lymphatic plexuses are found here in the superficial submesothelial space and the deeper lymphovascular bundles within the mesenteric connective tissue lattice [6,7,8]. From the mesenteric collecting lymphatics, the lymph is carried to the central/preterminal superior mesenteric nodes around the corresponding artery [2]. From there, the intestinal lymph or chyle (which also contains chylomicrons) is transported via the intestinal trunk(s) to the cisterna chyli and then through the thoracic duct into the left venous angle. While the peripheral [7,8,9] and the central [10,11,12] mesenteric lymphatics have been thoroughly studied, the link between them, i.e., immediate affluents of the intestinal trunk, is still scarce in the literature, even though they are of paramount clinical significance in gastric, colorectal, small bowel and pancreatic surgery, e.g., in cases of chylous ascites [13]. Further, it has been reported that lymph vessels draining the pancreas can open directly into the thoracic duct or via para-aortic lymph nodes [14]. The D3 lymphadenectomy implies removing as much lymphatic tissue as possible in the base of the mesentery, consequently leading to injuring of a higher proportion of the lymph vessels and/or larger lymph vessels, resulting in lymph leakage.

Throughout the development of lymphangiology, the classical anatomical dissection [6,7,14,15] remains one of its cornerstones. Minute dissection can present even more delicate lymph vessels and lymph nodes, as well as the meshes which they form [7,14,15]. On the other hand, the clinical visualization of the mesenteric lymph vessels faces difficulties. Although the mesenteric lymph (chyle) is milky in appearance, the system is hardly visible because of its fine plexuses. Therefore, the majority of interest in high-resolution imaging focuses on the blood vessels and larger organs [15]. Currently, the visualization of the lymphatic vessels is primarily achieved by direct lymphangiography, followed by MDCT [16], near-infrared indocyanine-green fluorescent imaging [4], on T2-weighted MRI sequences [10], or by dynamic contrast-enhanced magnetic resonance lymphangiography [17]. 

In the available literature, there seems to be a gap in knowledge of the anatomy of mesenteric lymphatics between the superior mesenteric nodes and the intestinal trunk. To our knowledge, these central lymph vessels were not hitherto systematically searched for, described, or morphometrically analyzed, particularly in a clinical setting. 

### 1.2. Aim

We aimed to present large mesenteric lymph vessels identified as additional findings in the D3 volume on MDCT angiography (MDCTA) prior to right colectomy for cancer, with extended mesenterectomy, central vascular ligation, and D3 lymphadenectomy. Further, those vessels should be thoroughly analyzed, according to their size, topography and morphology. 

## 2. Material and Methods

The pilot portion of this study is based on the anatomical material of a previous investigation of lymph vessels [8]. In short, the fresh human excisates containing the duodenopancreas and the surrounding small and large bowel and their mesenteries were explanted during the autopsy, thoroughly rinsed in running water, and then fixed in a 4% formol solution. Detailed stepwise microdissection of the D3 volume was carried out with the aid of the magnifier and adequate surgical instruments (scissors, clamps, forceps, tweezers, spatulas), starting from the ileocolic artery and proceeding cranially. After carefully removing the subserosal and deeper fatty tissue, the lymph vessels, plexuses and nodes were presented in a layer external to the superior mesenteric artery (SMA) nerve plexus, i.e., the SMA sheath. The course of the lymph vessels was followed in relation to the colic and jejunal arteries and main affluents of the superior mesenteric vein (gastrocolic trunk of Henle, right colic vein, ileocolic vein, jejunal veins, ev. middle colic vein).

The material for the central part of this study derives from the clinical trial ‘Safe Radical D3 Right Hemicolectomy for Cancer through Preoperative Biphasic Multidetector Computed Tomography Angiography’, which received the IRB approval REK Sør-Øst No. 2010/3354, Norway, and was recorded at clinicaltrials.gov (NCT01351714) [18]. A prerequisite for entering the study was a preoperative computerized multidetector computed tomography angiography of the abdomen and pelvis. The scanning parameters are given in Table 1. The Dicom datasets underwent minute image analysis through manual 3D reconstruction, presenting the complete mesenteric vascular tree–the superior mesenteric vessels and their branches and affluents. For that purpose, two software applications were applied: 1. FDA cleared Osirix MD v. 12.0.1 64-bit image processing package (Pixmeo, Bernex, Switzerland); and 2. Mimics medical image processing software, ver. 24.0, and 3-matic medical software, ver. 16.0, both two Windows 10 Pro x64 (Materialise NV, Leuven, Belgium). The manual segmentation in Osirix was obtained by serial application of Region of Interest (ROI) through editing tools: Open polygon, Pencil, and Repulsor; then, after setting pixel values outside ROIs to air, the virtual model was obtained by 3D volume rendering (VR). Likewise, the manual segmentation in Mimics was obtained via Profile line thresholding, Single and Multiple slab editing with interpolation, Dynamic region growing, 3D Interpolate, and Boolean operations. The final product of the imaging was a 3D VR model and stereolithography (STL) file. The morphometry of the vascular elements in the 3D file was carried out through 3-matic medical, using the Radius and Length (over the surface, true shortest path) tools. 

We carefully observed the wide D3 volume around the superior mesenteric artery and vein for the possible presence of lymph vessels. Only vascular structures without origin or termination on the SMA/V or their branches and affluents were considered in order to avoid confounding with eventual vascular shunts. If identified, the course of lymph vessels was analyzed concerning the surrounding vascular landmarks (SMA, ileocolic artery, middle colic artery), and their dimensions (diameter and length) were measured. 

The complete anatomical report, including video clips, still images, and STL files, was conveyed to the operating surgeon for making a “road mapping” during the surgical procedure. 

## 3. Results

The pilot project of this study was aimed at the lymph vessels situated distal (in the sense of lymph flow) to the lymphovascular bundles of the colic arteries, and the watershed lymph drainage area between the small and the large bowel. First, the lymphatic meshes were found superficial to the paravascular nerve sheaths of the colic arteries, and the SMA. Second, the dissection revealed a large longitudinal lymph vessel in the vicinity of the SMA, coursing through the D3 volume. It crossed the left-hand side of the middle colic artery (MCA), deep to its left branch, surrounded by a network of more superficial lymph vessels and nodes (Figure 1). Regarding the position and course of this larger lymph vessel, it can be attributed to the immediate affluents of the intestinal trunk.

A total of 420 patients entered the central portion of the study. The detailed anatomical setting of the superior mesenteric vessels, their affluents, and branches in the D3 volume, presented by segmentation and 3D reconstruction, was fully verified at the surgery. The significant vascular-like structures, having neither origin nor termination on the blood vessels, were noted in 18 cases (4.3%) in the D3 volume (Table 2). Their CT datasets had mainly slice thicknesses of 1 mm or below. The dimensions of visible lymph vessels varied, the mean diameter was 1.81 ± 0.61 mm (range 0.92–3.10 mm), and the mean length was 38.07 ± 22.19 mm (range 18.17–101.28 mm) (Figure 2). In the vast majority of cases, the lymph vessels were situated in front of the SMA, coursing either longitudinally cranially (13 cases) or transversely/obliquely to the left (5 cases, Figure 3). In the two most extended cases, the lymph vessel was visible beyond the D3 volume, passing dorsal to the spleno-mesenteric venous junction and pancreas. The level of depart of lymph vessels was leveled with MCA or just above it in 13 cases, while in 5 cases, the depart was situated below the MCA level. In all cases but one, the lymph vessel passed at the left-hand side of the MCA. The partial proximal relations to the superior mesenteric vein were registered in only three cases. As for the course shape, in 7 cases, the lymph vessel appeared as highly serpiginous (Figure 4); the remaining 11 cases had lymph vessels of a straight or mildly wavy course. 

## 4. Discussion

Our pilot dissection confirmed that the bowel lymph nodes and vessels lie outside the paravascular sheath of the SMA and its branches [19]. Further, it has precisely presented a large lymph vessel in the D3 volume as an additional finding. This discovery has initiated a search for such entities in our MDCTA series.

The Classification of Colorectal Carcinoma [19] identifies the main (lateral) lymph nodes from stations 203, 213 and 223 at the origins of the ileocolic, right colic and middle colic artery, respectively, and lymph nodes proximal to main lymph nodes from station 214, i.e., the superior mesenteric nodes. From that point onward, the fate of the mesenteric lymph is somewhat ambiguous. The Atlas of lymphography [20] presents a group of three central mesenteric nodes which give rise to two short lymphatic constituents of the intestinal trunk, coursing just above the SMA origin.

It is worth noting that the anatomical dissection reveals, apart from the regular lymphovascular bundles surrounding and following the colic vessels, the independent longitudinal lymph vessels, which cross the SMV and, without interruption, terminate in the main/central lymph nodes [7]. These give theoretical bases for the “skip metastases” and imply that the whole mesenteric tissue in front and behind both the SMA and the SMV should be excised during D3 colectomy.

The majority of the authors mention the intestinal trunk [1,10,11,16,21,22,23,24], but, on the other hand, Usovich et al. [25] found no classical lymphatic intestinal trunk in 130 anatomical specimens. Contrary to the latter, a later anatomical study on almost the same sample size [11] found that the left lumbar trunk most frequently joins the intestinal trunk in forming the cisterna chyli. The other variants were union with the right lumbar trunk, or simultaneous junction of the intestinal trunk, the two lumbar lymphatic trunks, lymph vessels from the retroaortic nodes, and lower intercostal collectors. 

The intestinal trunk was recognizable on magnetic resonance cholangio-pancreatography [10] in 13.6% of the cases and was detected in front of the cisterna chyli. On our MDCTA dataset, we had a 4.3% rate of identification of lymph vessels in the D3 volume. The methodology differences can explain this discrepancy—the standard MDCTA is primarily not designed for lymphography, and by the fact that the vessels we found were confined to the D3 volume, i.e., most probably the lymphatic affluents instead of the intestinal trunk itself. 

The classic monographs present data on the central intestinal lymph collectors [21,22]. Apart from a single intestinal trunk, it is not uncommon to find multiple efferent vessels of the superior mesenteric nodes. Occasionally, the most prominent among them is called the intestinal trunk. It courses downward and to the left, crossing the anterior aspect of the left renal artery, and, after skirting the left side of the abdominal aorta, joins the left lumbar trunk. The remainder of the efferent vessels empties in the left, rarely in the right aorticorenal lymph node. Servelle and Nogués [22] have identified four–six lymphatic trunks which cross the uppermost part of the small bowel mesentery, then descend in front of the left renal vein, go around its inferior border, then ascend to the right, opening into the cisterna chyli. Apart from this pattern, there are numerous anastomoses with the preaortic lumbar lymphatics. In our series, there was only one vessel identified per case with visible lymphatics. It coursed predominantly longitudinally along the superior mesenteric artery, crossing the left-hand side of the middle colic artery, which corresponds to the watershed between the small bowel and right colon lymphatics [7]. This fact is crucial for road mapping in complete mesocolic excision, as lesions of the larger lymph vessels can lead to lymph leakage [13].

We found that the visible lymph vessels were highly tortuous in seven cases. This variable appearance can be attributed to the patterning and remodeling of collecting lymphatic vessels from initial plexuses [3]. The development of lymphatic vessels is regulated by several factors, such as Foxc2, podoplanin, ephrinB2, angiopoietin-2, some of which can be used for their immunohistochemical identification [7].

Concerning the size of the intestinal trunk, the anatomical study of Ji et al. [23] measures it as 36 mm in length but does not give its diameter. Our study has carried out detailed morphometry on the visible lymphatics, giving a mean length/caliber of 38.07/1.81 mm, but with a wide range of variations. It is worth noting that the size of the mesenteric lymphatics depends upon numerous factors. For instance, the high dietary fat [13] and the upright position of the body [12] can significantly increase their diameter. The caliber of the lymph vessel is a measure of its flow capacity. The thoracic duct has a flow rate between 60 and 190 mL per hour [1]. As the input proportion of the small bowel in this flow is much more significant than the one of the large bowel, we can rather ascribe the identified large lymph vessels to the former than to the latter. This fact is even more fortified by the position and course of this vessel, as it predominantly lies to the left of the “watershed” between the small bowel and the right colon lymphatics, as defined by a postmortem study [7].

Within the framework of this study (right colectomy for cancer), there is yet another factor influencing the size of the lymph vessels. Namely, suppose the lymphatics around the origin of the SMA can be considered as a “bottleneck” for all the affluents draining the small bowel and the right colon. In that case, it is close to reason to assume that local obstruction of the lymph system by the lymph nodes attaint by metastasis would lead to a retrograde dilation of the more proximal lymph vessels, which therefore become more visible. The exact mechanism can be applied for malignancies situated in the proximity of the SMA, e.g., pancreatic cancer. This hypothesis requires clinical evaluation. Given the fact that the majority of incipient metastases are spread employing the lymphatic vessels [25], one question arises. Are the visible lymph vessels in our study part of the “normal” anatomy or a consequence of the lymphangiogenesis initiated by the malignant disease? It is known that angiogenic factors from group VEGF (A,C) can induce lymphangiogenesis, but this occurs primarily within the tumor and at the tumor margin [25]. Therefore, the central lymphatic vessels identified here should be ascribed to the pre-tumoral structures. Last but not least, the in vivo visualization of mesentery lymphatics in clinical settings has presented advantages and drawbacks. The MRI [10,12] presents the cisterna chyli principally, and the indocyanine green fluorescent imaging [4] gives the peripheral lymph vessel and the targeted node but not beyond it. On the other hand, the MDCT after direct lymphangiography can visualize details such as intestinal lymph leakage, intestinal trunk reflux, and abdominal lymph leakage [16], but it requires a pre-intervention which the oedema of the lower extremities can hamper. The dynamic contrast-enhanced MRI is based on injecting the gadolinium agent into the inguinal lymph nodes, followed by a maximum intensity projection of T1-weighted sequences [15]. This method gives a presentation of the thoracic duct and the cisterna chyli and is of value in detecting central lymphatic disorders, e.g., leakage in the thoracic cavity. However, the details of the pattern of the more proximal lymph vessels are limited. Our MDCTA series has shown vessels in the D3 volume, which can be attributed to the lymphatic system. Most probably, their appearance is due to transvasation of the contrast media from blood to lymph vessels. Their detailed visualization has enabled a thorough topographic and morphometric analysis. 

As already mentioned, the complete mesocolic excision implies a higher yield of lymphatic tissue and, ultimately a better patient outcome [5]. It depends on the preoperative road-mapping of vascular structures in the D3 volume, based on the 3D reconstruction of the mesenteric blood vessels. Herewith, we introduce the visualization of the lymph vessels as a significant addition to image analysis and a valuable aid in surgical planning.

Concluding, our study has shown that routine MDCTA prior to surgery can also present essential lymph vessels in the region of interest without the use of invasive methods, in contrast to the usual approach to mesenteric lymph pathology. Additionally, we used the CT datasets for a detailed 3D reconstruction, morphological, topographical and morphometric analysis of the intestinal trunk affluents in the D3 volume, which has not been reported hitherto. 

## Figures and Tables

**Figure 1 diagnostics-12-02441-f001:**
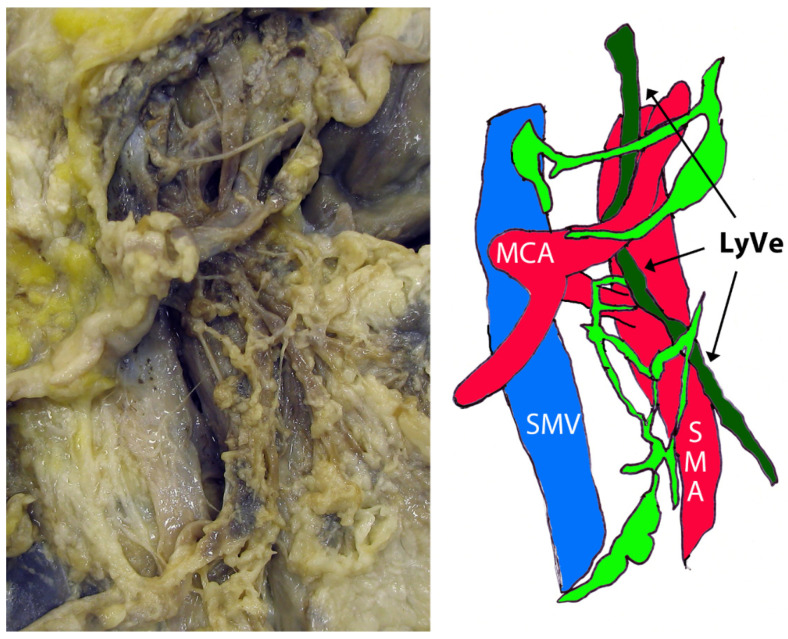
Dissection of the D3 volume and schematic presentation. The middle colic artery (MCA) is seen with its bifurcation. SMV—superior mesenteric vein; SMA—superior mesenteric artery; LyVe—large lymphatic vessel; light green—superficial lymph vessels and nodes.

**Figure 2 diagnostics-12-02441-f002:**
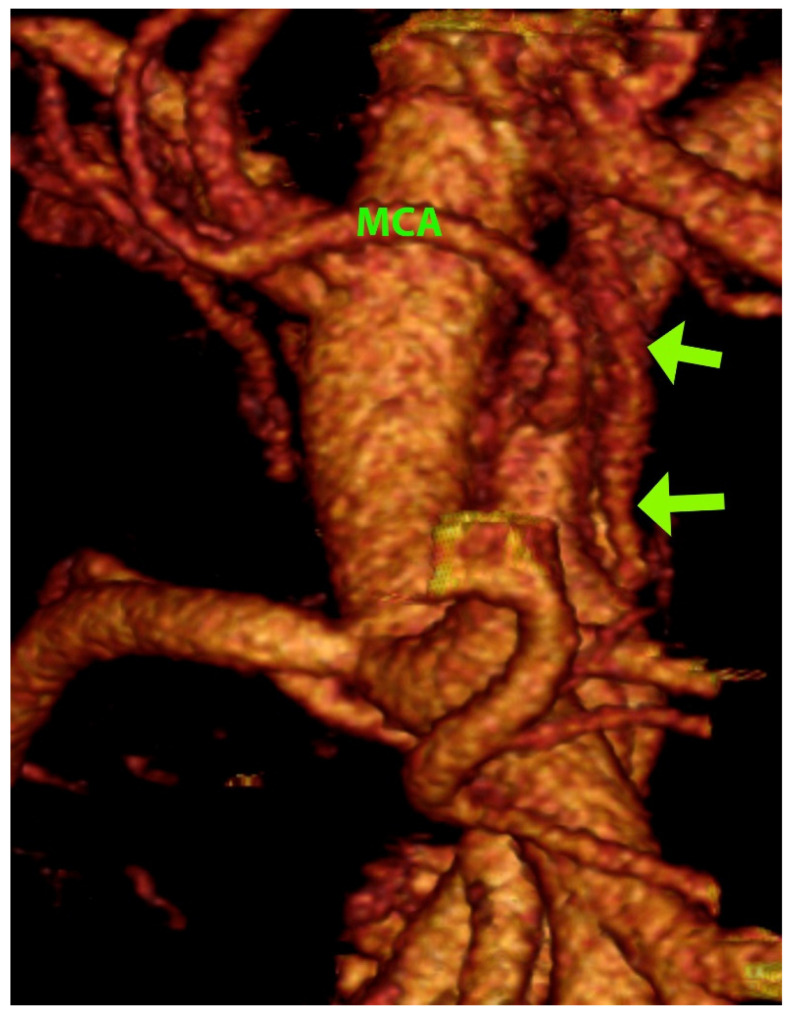
Three-dimensional volume rendering of the D3 volume. MCA—middle colic artery, arrows—largest lymph vessel found (caliber 3.10 mm).

**Figure 3 diagnostics-12-02441-f003:**
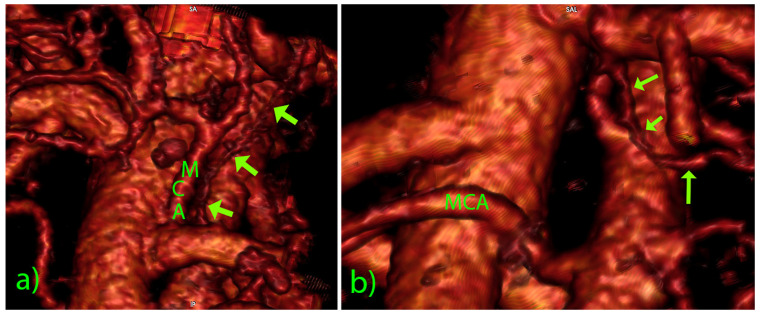
Three-dimensional volume rendering of lymph vessels (arrows): (**a**) longitudinal vessel paralleling the MCA; (**b**) oblique course of the lymph vessel.

**Figure 4 diagnostics-12-02441-f004:**
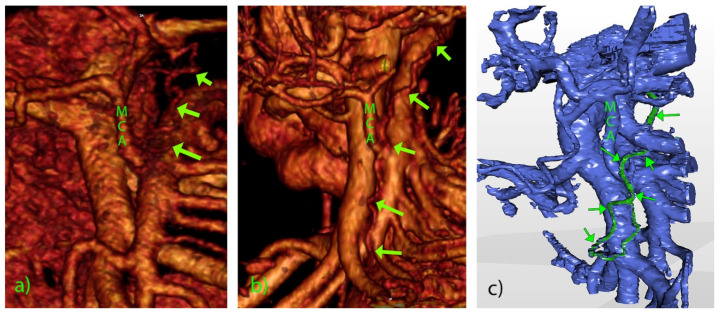
Three-dimensional volume rendering (**a**,**b**) and STL (**c**) of the D3 volume. Captions as in Figure 2 and Figure 3. Serpiginous vessels (arrows) coursing from the MCA level (**a**) or the level of the ileocolic vessels (**b**,**c**).

**Table 1 diagnostics-12-02441-t001:** The scanning parameters of the MDCTA.

Voltage 120 kV
Tube current 440–747 mA
Single collimation 0.6–3.0 mm
scan-pitch ratio 1.375:1; reconstruction interval 0.452 mm.

**Table 2 diagnostics-12-02441-t002:** Morphology and morphometry of visible lymph vessels in the D3 volume.

Case	Sex	Age	Caliber (mm)	Length (mm)	Course (Related to the SMA)	Level (Related to the MCA)	CT Slice (mm)	Serpiginous
1	M	60	1.60	26.36	in front, longitudinal	From ICA, left to MCA	3	No
2	F	63	1.68	28.68	in front, longitudinal	From MCA, left to MCA	3	No
3	M	74	2.20	52.69	in front, longitudinal	From MCA, left to MCA	0.9	Yes
4	M	64	3.10	33.09	to the left, longitudinal	From MCA/ICA midpoint, left to MCA	0.67	No
5	M	68	2.93	24.85	in front, longitudinal	From GTH, right to MCA	3	Yes
6	M	75	1.20	21.21	in front, transverse	Above MCA, left to MCA	0.9	No
7	M	75	1.91	30.23	in front, transverse	Above MCA, left to MCA	2	Yes
8	F	46	1.93	24.16	in front, transverse	Above MCA, left to MCA	0.9	No
9	M	72	0.92	18.17	in front, transverse	Above MCA, left to MCA	0.9	No
10	M	60	1.39	3.663	in front, longitudinal	From MCA, left to MCA	2	No
11	M	51	1.20	2.985	in front, longitudinal	From MCA, left to MCA	0.9	No
12	M	77	1.96	8.134	in front, longitudinal	From ICA, left to MCA	1	Yes
13	F	74	1.81	3.636	in front, longitudinal	From MCA, above MCA	1	No
14	M	55	2.75	3.274	in front, longitudinal	From MCA, above MCA	1	Yes
15	F	66	1.57	2.812	in front, longitudinal	From RCA, left to MCA	1	No
16	M	61	1.60	5.879	in front, longitudinal	From ICA, left to MCA	1	Yes
17	M	66	1.07	2.079	in front, oblique	Above MCA, left to MCA	1	No
18	F	64	1.82	10.128	to the left, longitudinal	From MCA, left to MCA	0.9	Yes

## Data Availability

The datasets used and/or analysed during the current study are available from the corresponding author on reasonable request.

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
