# Peer review of "Visible Lymph Affluents in the D3 Volume: An MDCTA Pictorial Essay"

_diagnostics, 2022, doi:10.3390/diagnostics12102441_

Round 1
Reviewer 1 Report
First of all I would like to congratulate you the topic which seems to be original and clinically important.
I have few minor remarks which may be useful to improve the quality of your paper.
1. Studies on the lymphatic system have been undertaken long long time ago - I fully understand your posiotion to cite references from XXI century mostly but apart from the big monograph by Ludwik Karol Teichmann "Das Saugadersystem" Leipzig 1861- his excellent specimens are still preserved in the historical museum of the Anatomy Department of the Jagiellonian University Medical College in Kraków/ Poland.
Ludwik Karol Teichmann. Das Saugadersystem. Leipzig. 1861 (in German)
(which made him world famous and earned him the title of the "last gross anatomist" in the world" it is worth to mention other reviews on the lymphatic anatomy which were performed not so long time ago:
The surgical anatomy of the lymphatic system of the pancreas
Alper Cesmebasi, Jason Malefant, Swetal D. Patel, Maira Du Plessis, Sarah Renna, R. Shane Tubbs, Marios Loukas
Clinical Anatomy Volume 28, Issue 4
First published: 15 September 2014
The lymphatic system: A historical perspective
, , , , , , ,
Clinical Anatomy Volume 24, Issue 7
The lymphatic system throughout history: From hieroglyphic translations to state of the art radiological techniques
, , , , , , ,
Clinical Anatomy Volume 35, Issue 6
Translational Research in Anatomy
Volume 23, June 2021, 100105
Anatomical normality and variability: Historical perspective and methodological considerations
, R. Shane Tubbs, Joe Iwanaga, Edward Clarke, Michał Polguj, Grzegorz Wysiadecki
https://doi.org/10.1016/j.tria.2020.100105
My main concern is associated with the material used: is this really "anatomical"? My doubts are associated with the fact of proved lymphogenesis which may be initiated by cancer. I think that tumor development may impinge lymphatics and change their position and course. Obviously the study is very important but I would be sceptical if this is norm...
Reviewer 2 Report
This paper presents a study on the "Visible lymph affluents in the D3 volume. A MDCTA pictorial essay". This is an interesting manuscript for this journal but I suggest a minor revision. Here are some bugs in this article to help the authors to profit from this article, but if the authors can't do these comments (point by point) the article will be rejected.
================================
1) General comments:
1a) The English language is poor and should be enhanced. Please take time to improve the language. Its current version is poor.
1b) Discussion is not enough. Authors should add some technical description to the manuscript (major comment).
================================
2) Keywords:
2a) The authors must update keywords. These keywords don't cover this article.
================================
3) Abstract:
3a) The abstract doesn’t have novelty in it. The authors should rewrite the abstract with main novelty in it.
3b) What is the main purpose of the article? The authors should focus on novelty on this section.
================================
4) Introduction and Literature Review:
4a) The introduction is very brief. The authors should extend it (some material about and novelty).
4b) I strongly recommend the authors add a new headline (1.1. literature review). At least, 6 literature review is required with more detail and their novelties (major comment).
4c) At the end of this section, the novelty of the article should be mentioned and the difference between this article and the articles they researched in this field (major comment).
================================
5) Material and Methods:
5a) I didn't see any configuration of model. How authors confirm this model?
5b) The main parameters for this method must input as table in this article.
================================
6) Results:
6a) The authors should add some technical description to this section.
6b) Result is not enough to description of the main challenge..
================================
7) Conclusions:
7a) The authors should mention the novelty of the article and the novelty of the technique.
7b) This section should be completely rewritten and should be written as one or two paragraphs in which all the results of the article and also the difference between this article and other articles (without any numbers).
================================
8) References:
7a) References are very old, and I strongly suggested the author’s update references
Round 2
Reviewer 1 Report
for me ok.